# The frequency, risk factors and spatial distribution associated with having a diagnosis of leishmaniosis in dogs in the UK in 2019

**Simon C. Kent**[1]*, **Dan G. O'Neill**[1], **Kim B. Stevens**[1], **Laia Solano-Gallego**[2], **Dave C. Brodbelt**[1]

1 Pathobiology and Population Sciences, The Royal Veterinary College, Hatfield, Herts, United Kingdom,
2 Departament de Medicina i Cirurgia Animals, Universitat Autònoma de Barcelona, Barcelona, Spain

* skent23@rvc.ac.uk

## Abstract

### Background

Leishmaniosis in dogs is an important zoonotic disease, endemic in southern Europe but is increasingly being recorded in other parts of Europe including the UK. This study aimed to update the epidemiological knowledge of leishmaniosis in the UK dog population.

### Methods

The estimated prevalence of leishmaniosis diagnosis in dogs under care at VetCompass UK participating practices in 2019 was calculated. Risk factor associations for leishmaniosis were investigated using mixed-effects multivariable logistic regression modelling. The spatial distribution of leishmaniosis was explored using standardised morbidity rates (SMR) at regional and county levels, and Moran's I was used to identify significant spatial autocorrelation.

### Results

The estimated annual prevalence was calculated as 0.0434% (95% CI 0.0427–0.0443) with 976 cases from this population of dogs (n = 2,250,741). Most of the dogs with leishmaniosis had originated from outside the UK (955/976, 97.9%) and of those, 60.8% (581) had already been diagnosed with leishmaniosis pre-import. In the risk factor analysis, the odds of leishmaniosis were higher in males (OR 1.29, 95% CI 1.12–1.49) and neutered dogs (OR 6.42, 95% CI 4.43–9.29) compared to female and entire dogs, respectively. Purebreds overall had lower odds of leishmaniosis compared to crossbreeds (OR 0.24, 95% CI 0.20–0.28) but there were specific breeds such as Brittany Spaniels (OR 53.0, 95% CI 25.3–111.1), Ibizan Hounds (OR 43.8, 95% CI 9.20–208.3) and English Pointers (OR 28.6, 95% CI 14.5–56.4) which had

**Data availability statement:** The anonymised datasets have been uploaded to Figshare https://doi.org/10.6084/m9.figshare.30100243.

**Funding:** The author(s) received no specific funding for this work.

**Competing interests:** The authors have declared that no competing interests exist.

much higher odds of being a case compared to crossbreeds. Spatial analysis identified more leishmaniosis cases in southern England than were expected by random chance (SMR > 1).

## Conclusions

This study identifies that imported animals are the commonest source of dogs having a leishmaniosis diagnosis. The breeds with very high odds for leishmaniosis may reflect selection bias towards breeds more likely to be imported, rescued or travelled from endemic areas rather than high intrinsic risk in these breeds towards leishmaniosis. These results support and update previous studies indicating that most cases are diagnosed in southern England.

---

## Introduction

Leishmaniosis in Europe is predominately caused by the protozoal parasite *Leishmania infantum* and spread by *Phlebotomus* sandflies [1,2]. Leishmaniosis is a zoonotic disease, with canid species acting as the major reservoir of infection [1–3] and reported to cause approximately 700 human cases a year in southern Europe alone [4]. In Europe, leishmaniosis is endemic in the Mediterranean basin [2,4–7]. There is concern that leishmaniosis will spread northwards as climate changes extend the habitat range for the sandfly vector [8,9]. In the UK, there is also concern that the frequency of leishmaniosis diagnosis in dogs will rise due to increased travel and importation of dogs from endemic areas [10]. In 2021, there were 72,766 commercial dog imports and 57,411 dogs rescued from the EU. Clinical leishmaniosis in dogs is a serious and incurable condition often causing high levels of morbidity and mortality [11,12]. Transmission risk in the UK remains very low because no incursions of the sandfly vector have been recorded into the UK. Non vector-borne transmission from infected canine blood banks, venereal and vertical transmission have been recorded [13–15] with only one case report suggesting vertical transmission possibly occurred in the UK [16]. Additionally, one case report from the UK has suggested that dog to dog transmission through bites and wounds has occurred [17].

In endemic areas, after infection with *L. Infantum,* often less than 10% of dogs will develop clinical disease and the rest will be subclinical [18]. Formal diagnosis of clinical leishmaniosis can be challenging and is recommended to be based on cumulative evidence from typical clinical signs and/or routine laboratory abnormalities combined with laboratory test detection of parasite or detection of antibodies against *L. infantum* antigen [11,12,18,19]. Underestimation of clinical leishmaniosis in dogs is possible by clinicians, especially in non-endemic countries, as there is a wide spectrum of clinical signs and/or clinicopathological abnormalities with none being pathognomonic [19–21]. It can be challenging for clinicians unfamiliar with the disease, to achieve a high enough index of suspicion to proceed to laboratory diagnosis [19]. The majority of dogs with subclinical leishmaniosis will remain infected but not develop clinical disease. The remainder will develop disease after a variable period

of time depending on the dog's immune response [8,18,20,22]. Subclinical dogs remain important in prevalence estimation as they may become clinical cases and in endemic countries, possibly act as a reservoir of infection for sandflies to spread disease [8,20]. Seropositivity is found in 88–100% of dogs with clinical signs and/or clinicopathological changes consistent with leishmaniosis, but is found in less than 30% of clinically healthy infected dogs [18]. Therefore serological tests are less useful in differentiating non-infected dogs from dogs with subclinical leishmaniosis. This is further complicated by imperfect serological tests with less than 100% specificity and sensitivity [23–28]. These factors can complicate prevalence estimation of leishmaniosis.

Prevalence risks of leishmaniosis are a product of incidence risk of infection and duration of disease, therefore for chronic diseases like leishmaniosis it can be difficult to separate factors that are associated with survival from those associated with cause but are nonetheless relevant as a measure of the disease burden in the population [29]. Reported prevalence risk in dogs calculated by the percentage of practice-attending dogs with a confirmed leishmaniosis diagnosis has been estimated to be as high as 7.8% in Greece [5]. One previous study from the UK reported a prevalence estimation of between 0.007–0.04% though this related to 38 cases from a referral centre caseload from 2005–2014 [19].

Risk factors such as being male [30–32], advancing age [2,30,31,33–36], purebreds compared to crossbreds [2,31,37] have all been found to have positive associations with leishmaniosis diagnosis in dogs, although there is little consensus between studies. A systematic review of risk factors associated with leishmaniosis in Brazil suggested that this variation between studies reflected the use of differing case definitions and diagnostic tests [37].

Previous UK studies have found increased numbers of cases in southern England [10,19,38]. In 2009, a UK study looking at dogs diagnosed at a referral centre laboratory or diagnosed at other laboratories that had requested advice from the referral centre laboratory, found more cases in the south of England although this was a study of laboratory submissions and may have reflected that there were more samples from this region [10]. A recent UK study looking at electronic health records (EHR) of dogs attending 251 veterinary practices between March 2014 and September 2022 within the Small Animal Veterinary Surveillance Network (SAVSNET) also found more cases in the southeast of England [38]. This same study used a retrospective case-control study to determine risk factors for leishmaniosis and found that cases were more likely to be male compared to female, neutered compared to entire, and aged between three and six years old compared to under two years old. Breed analysis showed that Pointers and crossbreed dogs were more likely to be a case than Retrievers, although individual breed types were not evaluated. The current study aimed to explore further the epidemiology of leishmaniosis within dogs under care at VetCompass participating practices. Emphasis was placed on updating the estimated prevalence, exploring individual breed associations and developing further the spatial analysis of the risk of having a diagnosis of leishmaniosis. It was hypothesised that, consistent with previous UK studies, cases would be more likely in the south of England

## Materials and methods

The study population included dogs under primary veterinary care from participating UK practices in VetCompass in 2019. Dogs under veterinary care were defined as those with an electronic health record (EHR) recorded in 2019. Data available from the EHR in VetCompass included a unique identification number as well as species, breed, date of birth, sex, final neuter status, clinic id, practice postcode and free-form text, clinical notes, and treatment with relevant dates. Ethics approval was obtained from the Royal Veterinary College Ethics and Welfare Committee (reference SR2018−1652).

### Case definition and disease frequency estimation

A case-control study within the cohort of dogs under veterinary care within VetCompass practices during 2019 was used to estimate a one-year period prevalence of leishmaniosis and to explore associations with a range of risk factors.

Dogs were defined as cases if they had evidence in their clinical records of having clinical or subclinical leishmaniosis. Clinical leishmaniosis was diagnosed when the dog had presented (on or before 31/12/2019) with typical clinical signs

(including skin lesions, weight loss, uveitis, polyarthritis, anaemia, renal disease) and/or typical clinicopathological abnormalities [39]. The positive diagnosis was confirmed by a commercial UK laboratory although the specific laboratory or diagnostic facilities where each test was performed were not available to the study. The results were interpreted according to that individual laboratory's reference range. Results were recorded as either positive or negative as numerical values for each result could not be compared between laboratories. The number of positive cases that were confirmed with more than one test result was not extracted for this study. Diagnostic test confirmation was either serological with immunofluorescence assay (IFAT) or an enzyme-linked immunosorbent assay (ELISA) and/or via detection of *Leishmania* DNA by polymerase chain reaction (PCR or qPCR) from blood, lymph node, or conjunctiva samples. Dogs were additionally classified as cases, if they had exhibited no typical clinical signs by 31/12/2019 but then developed clinical disease by 30/05/2024 (end of the data collection period) with the presence of a confirmatory test as described above. Subclinical leishmaniosis cases were defined as dogs that remained free of clinical signs but had persistent high seropositivity or presence of parasite in laboratory tests. The latter group were usually imported dogs that were being regularly monitored by blood testing. Dogs diagnosed with leishmaniosis pre-import, that remained on treatment without further confirmatory laboratory tests after arrival in the UK, were additionally classified as cases. Dogs that had been diagnosed as cases from laboratory tests pre-import, that remained free of clinical signs and subsequently tested negative on confirmatory laboratory tests in the UK by 01/01/2019, were considered exposed to the parasite but not infected and were not considered cases for the current study. All levels of parasite detection by PCR from blood, lymph nodes or conjunctiva samples were considered positive if dogs showed clinical signs typical of the disease.

Candidate leishmaniosis cases were identified from the 2,250,741 dogs that had at least one EHR recorded between 01/01/2019 and 31/12/2019 using the search term leish* in the treatment and clinical text fields. Candidate cases were manually evaluated against the criteria for the case definition and classified as either cases or non-cases. Information extracted for all cases (if available) included date of initial diagnosis, diagnostic tests used, disease progression, date of dog's death or date last seen alive, treatment history, country of origin if imported or countries visited if there was a history of international travel. The clinical histories for all candidate cases were followed from their first EHR up to 30/05/2024.

Using the estimate for the 2019 UK dog population as twelve million, a prevalence of 0.04% which was the maximum estimate from the only UK study prevalence estimate [29], a precision of 0.01% and a 95% confidence limit, a required study sample of at least 151,662 dogs was estimated (Epitools Epidemiological Calculators).

### Risk factor analysis

A prevalent case risk factor analysis was undertaken as most dogs had been infected before import and therefore, before they entered the overall study population. The risk factor analysis population included dogs aged ≥ one year on 01/01/2019 in both cases and controls. This age restriction was chosen to allow all dogs a minimum period to become confirmed cases. Controls were randomly selected from a list of unique animal identification numbers for all dogs with an EHR recorded between 01/01/2019 and 31/12/2019 that did not have reference to the search term leish* in their EHR.

Breed information was standardised to VeNom breed terms [40]. Recoding was performed where there were repeated terms for the same breed (such as French Brittany Spaniels and Brittany Spaniels). Recoding of the cross-breed breeds such as Cockerpoo to crossbreed was performed. A breed variable was developed based on cross-breeds, several individual breeds and the rest categorised as 'purebred other'. The risk factor study analysed sixteen individual breeds which included those breeds with six or more dogs having a leishmaniosis diagnosis, with all other purebred dogs grouped together in a separate 'purebred other' category and crossbreds grouped separately as an additional group. A further breed-derived variable was created and called purebred status/coat length by assigning a coat length based on the expected phenotypic characteristic of each breed. Crossbreed was included as a separate group as estimating coat length was considered impossible. The last recorded neuter status was also evaluated for risk factor analysis.

Age was defined as the age on 01/01/2019 for both cases and controls. Age was assessed as a continuous variable and reported as median and interquartile range (IQR) and explored also as a categorical variable divided into five groups. Median age of cases and controls was compared using a Wilcoxon-Mann-Whitney test. An Indices of Multiple Deprivation (IMD), country, NUTS 1 regions (Nomenclature des Unites territoriales statisques), county and urban development classification groups (4 category, see below) were allocated from the clinic full postcodes as additional variables in the risk factor analysis. IMD rank was evaluated by assessing seven domains of deprivation for each area (income, employment, crime, education, health and disability, housing and living environment). Data were collected by the Consumer Data Research Centre, and these were combined with the clinic's full postcode to separate each clinic address into five levels, with 1 as the most deprived and 5 as least deprived. Each postcode of the veterinary clinic visited was assigned to an urbanisation classification that was harmonised across all regions of the UK. This consisted of mixed urban/rural, rural, urban and the most developed category, urban city/town. The 'practice data source' represented the veterinary group to which the practice belonged to.

Data were exported to a spreadsheet (Excel for Mac, Microsoft Corporation). The data were cleaned, removing any duplicate observations and the clinical free text was used to manually check on any conflicting information. The data were exported into Stata v18 (StataCorp) for statistical analysis.

Mixed-effects logistic regression modelling was used to identify demographic and geographical risk factors associated with dogs having leishmaniosis. In both univariable and multivariable analysis, clinic ID was included as a random effect to account for clustering at the clinic level and the practice data source was included as a fixed effect and retained *a priori* in the multivariable analysis to adjust additionally for clustering at the practice group level.

From the univariable analysis, risk factors for having leishmaniosis were evaluated using a likelihood ratio test (LRT), with any factors having a liberal association ($p < 0.2$) being carried forward to the multivariable analysis. Missing observations among the risk factors were categorised as "missing" and assessed as a separate category and presented in the univariable analyses for clarity.

Variables taken forward for evaluation in the multivariable model were assessed for their collinearity via the chi-squared test. Breed-derived variables such as purebred status/coat length were excluded from the initial breed multivariable modelling. Purebred status/coat length then replaced the individual breed variable to evaluate their effects after adjusting for the other variables in the mixed-effects multivariable model. The final mixed-effects multivariable model was constructed using a manual backwards elimination approach. Confounding was assessed by the change in crude and adjusted odds ratios and considered if the difference was over 20%. Biologically significant interactions between age, sex and neuter status were assessed in the model using the LRT. Assessing whether age band as a categorical variable was a better fit than a linear age effect was carried out for each model with the LRT. The magnitude of clustering of the random effect was measured by the intraclass correlation coefficient (ICC). The area under the ROC curve was used to assess model discrimination. Statistical significance was set as $p < 0.05$.

**Spatial Analysis**

ArcGIS 10.8.2 (ESRI, Redlands, Ca, USA) was used for the creation of all maps and spatial analyses. Choropleth maps showing dogs having leishmaniosis standardised morbidity rates (SMR) and associated standard errors (se-SMR) per region or county of the UK, were created. SMRs were calculated using equation (1) where $Y_i$ is the SMR, $O_i$ is the region and county-level observed number cases and $E_i$ is the region and county-level expected number of cases. The se-SMR were calculated using equation (2), where $X_i$ is the se-SMR.

$$Y_i = [O_i/E_i] \tag{1}$$

$$X_i = [(\sqrt{O_i})/E_i] \tag{2}$$

The presence, strength and direction of region and county-level SMR spatial autocorrelation were assessed using Moran's I statistic with a queen's contiguity weight matrix and 999 permutations. Local spatial autocorrelation (LISA) using the same contiguity matrix was used to detect significant region/county-level local spatial autocorrelation and LISA maps generated to show the location of clustered and outlier counties.

## Results

### Prevalence of leishmaniosis cases

Text searches of the EHR of the 2,250,741 study dogs identified 9,116 (0.41%) candidate cases. All candidate cases were reviewed manually (by SK), and 976 cases met the case definition (10.7%). The one-year 2019 estimated prevalence for leishmaniosis diagnosis from this population was 0.0434% (95% CI, 0.0427–0.0443). Most of the cases had been rescued from overseas or imported (955, 97.8%). The rest had travelled from the UK to an endemic area outside the UK (20, 2.1%), apart from one dog which had no record of having ever travelled outside the UK (1, 0.1%). There was information on source country for 766 (78.5%) of the overseas rescued/imported cases. The highest number of imported cases came from Spain (414, 42.3%) with sizable contributions from Greece (159, 16.3%) and Cyprus (115, 11.8%). Nearly one-fifth of imported cases had no country of origin reported in the clinical records (189, 19.4%). Most imported cases had been diagnosed with leishmaniosis pre-import (581, 60.8%). The remaining cases were first diagnosed with leishmaniosis after they arrived in the UK (374, 39.2%) of which 54 (13.7%), had tested negative pre-import. The commonest clinical signs of the cases were cutaneous and mucocutaneous lesions (417, 42.8%). Many dogs were subclinical but qualified as cases by having persistent high seropositivity or presence of parasite (238, 24.4%). Treatment was given to most cases, (912, 93.4%) of which half were given allopurinol as a monotherapy (476, 52.3%). Three hundred and eleven cases (31.9%) had a recorded death by 30/05/2024 from which 113 (36.3%) deaths were considered associated with leishmaniosis based on available evidence (Tables 1 and 2).

### Risk factors for leishmaniosis

After age restriction to dogs one year and over, the risk factor analysis population included 915 case dogs and 38,812 randomly selected controls (cases: controls 1:42). In the cases, 58% were male (532) and 77% neutered (707). Median age of the cases on 01/01/2019 was 5.08 years (IQR: 3.32–7.71). The highest proportion of cases were crossbreeds (508, 55.5%). The most common breeds within the cases were Labrador Retriever and German Shepherd Dog (both 29, 3.2%). Within the purebred status/coat length variable, short haired purebreds were the most frequent purebred category (224, 24.5%). Most of the cases had attended veterinary clinics in England (860, 94.0%) and clinics that were classified as being in mixed urban/rural locations (567, 62.0%). The lowest proportion of cases attended veterinary clinics in the most deprived area (IMD category 1, 109, 11.9%).

Of the controls with data available, 51.7% were male (20,051) and 53.3% were neutered (20,690). The controls were slightly older than the cases with a median age of 5.74 (IQR: 3.24–9.03, p < 0.01). The commonest breeds in the controls were Labrador Retriever (2,713, 7.0%), Jack Russell Terrier (1969, 5.1%) and Staffordshire Bull Terrier (1811, 4.7%). Short-haired purebreds were the most frequent category within the purebred status/coat length variable (14,708, 37.9%). The smallest proportion of dogs in the control group attended veterinary clinics in least deprived areas (IMD 5, 6,721, 17.3%) although there was a more even spread within the five IMD categories (Tables 3 and 4).

All risk factors assessed by univariable mixed-effects logistic regression modelling were liberally associated with leishmaniosis and were evaluated using multivariable mixed-effects logistic regression modelling. The final breed-focused multivariable model retained eight risk factors: sex, neuter status, age, breed, country, IMD5 category, urban/rural 4-category and practice data source. Based on the LRT, the model was improved by including age as a categorical variable (p < 0.01).

**Table 1. Description of leishmaniosis cases including clinical presentation, treatment and confirmatory diagnostic tests performed from VetCompass data (n = 976 cases).**

| | Leishmanio-sis cases (%) | | Leishmanio-sis cases (%) |
|---|---|---|---|
| **Initial clinical presentation** | n = 976 | **Time of diagnosis** | n = 976 |
| Cutaneous/mucocutaneous lesions | 417 (42.8) | Pre-arrival in UK | 581 (59.5) |
| Renal disease | 77 (24.4) | In UK | 395 (40.5) |
| Polyarthritis/pain | 50 (5.1) | | |
| Pyrexia/anorexia/lethargy | 49 (5.0) | **Confirmatory test performed in UK** | n = 976 |
| Bleeding disorders | 48 (4.9) | Yes | 727 (74.5) |
| Ophthalmic disease | 41 (4.2) | No | 246 (25.2) |
| Weight loss | 32 (3.3) | Not recorded | 3 (0.3) |
| Crystalluria | 14 (1.4) | | |
| Other | 10 (1.0) | **Number of cases having at least one diagnostic test** | |
| Subclinical | 238 (24.4) | IFAT | 502 |
| | | ELISA | 249 |
| **Dogs having treatment** | n = 976 | PCR | 163 |
| Yes | 912 (93.4) | Unrecorded | 3 |
| No | 64 (6.6) | | |
| | | **Death recorded/last seen alive by 30/05/2024 in electronic health record** | n = 976 |
| **Treatment options** | n = 912 | Death | 311 (31.9) |
| Allopurinol only | 476 (52.3) | Last seen alive | 665 (68.1) |
| Domperidone only | 2 (0.2) | | |
| Miltefosine only | 4 (0.4) | **Attributed cause of death** | n = 311 |
| Allopurinol + Miltefosine | 242 (26.5) | | |
| Allopurinol + Meglumine antimonate | 59 (6.5) | Complication of leishmaniosis | 113 (36.3) |
| Allopurinol + Domperidone | 46 (5.0) | Not a complication of leishmaniosis | 198 (63.7) |
| Three or more drugs | 83 (9.1) | | |

The final model had an intraclass correlation coefficient (ICC) of 0.10 (95% CI 0.07–0.15) indicating that after adjusting for all variables in the model there was limited clustering of cases within a clinic. Model discrimination ability was good (AUC = 0.81, 95% CI 0.79–0.82).

After accounting for the risk factors included in the final model, male dogs had higher odds of leishmaniosis than females (OR 1.29, 95% CI 1.12–1.49). Neutered dogs had higher odds of leishmaniosis than entire animals (OR 6.42, 95% CI 4.43–9.29). Compared to dogs between one and three years, the odds of leishmaniosis increased with increasing age categories. There was an interaction between neuter status and age with the association of neutering with leishmaniosis reducing as the dogs increased in age. Breeds with the highest odds of being a case were Brittany Spaniel (OR 53.0, 95% CI 25.3–111.11), Ibizan Hound (OR 43.8, 95% CI 9.20–208.3) and English Pointer (OR 28.6, 95% CI 14.50–56.4) compared to crossbreds. Attending a veterinary clinic in Wales compared to England (OR 0.56, 95% CI 0.33–0.94), attending a veterinary clinic in a least affluent compared to most affluent area (OR 0.68, 95% CI 0.51–0.90) and veterinary clinics based in a rural compared to a mixed urban/rural location (OR 0.72, 95% CI 0.54–0.94) were all associated with reduced odds of being a case (Fig 1).

As described in the methods, a purebred status/coat length variable was introduced to replace individual breeds in the final breed-focused model. All purebred dogs whether they had short (OR 0.38, 95% CI 0.33–0.45), medium (OR 0.43,

**Table 2. Description of leishmaniosis cases including country of origin or from where travelled from Vet Compass data (n = 976 cases).**

| Country of origin | Leishmaniosis cases (%) n = 976 | UK origin dogs with a history of travel | Leishmaniosis cases (%) n = 21 |
|---|---|---|---|
| Spain | 414 (42.3) | Spain | 6 (28.5) |
| Greece | 159 (16.3) | Italy | 1 (4.8) |
| Cyprus | 115 (11.8) | Greece | 1 (4.8) |
| Portugal | 23 (2.4) | France | 1 (4.8) |
| Italy | 22 (2.3) | Portugal | 1 (4.8) |
| UK | 21 (2.2) | Multiple countries | 4 (19.0) |
| Turkey | 8 (0.8) | Unrecorded | 6 (28.5) |
| Romania | 6 (0.6) | Never travelled | 1 (4.8) |
| Tunisia | 5 (0.5) | | |
| France | 4 (0.4) | | |
| Morocco | 3 (0.3) | | |
| Bulgaria | 1 (0.1) | | |
| China | 1 (0.1) | | |
| Hungary | 1 (0.1) | | |
| Malaysia | 1 (0.1) | | |
| Montenegro | 1 (0.1) | | |
| Slovenia | 1 (0.1) | | |
| South Africa | 1 (0.1) | | |
| Not recorded | 189 (19.4) | | |

95% CI 0.35–0.51) or long (OR 0.13, 95% CI 0.08–0.21) haired coat phenotype had reduced odds of leishmaniosis compared to crossbred dogs.

## Spatial analysis

The highest SMRs of being a case were found in southern England. In the regional analysis, South East (1.60) had the highest SMR, with East (1.34), London (1.31) and South West (1.22) all having SMRs above one suggesting a higher than expected number of cases. Northern Ireland (0.31) and Yorkshire and Humberside (0.51) had half or under the number of expected cases (Fig 2).

These results were supported by the county analysis with East Sussex (SMR 2.90) and West Sussex (SMR 2.48) having the largest SMRs and twenty of the twenty-five counties with an SMR > 1, were from southern England. Tayside (2.67) and Clwyd (1.59) were outliers with much higher than expected numbers of cases but these were associated with high standard errors (1.20 and 0.65 respectively) reflecting the low number of dogs sampled from these areas (Fig 3).

There was significant positive spatial autocorrelation of regional and county-level SMRs in the UK with a global Moran's index of 0.447 (p < 0.01) and 0.446 (p < 0.01) respectively. Cluster mapping was performed at county level as there was increased statistical power analysing the 60 counties compared to the 12 regions. The LISA map showed significant low-low county clusters in the west of Wales and the border between England and Scotland. There was one significant high-low cluster in the south of England consisting of Hampshire, Surrey, West Sussex and East Sussex which all were high SMR counties surrounded by lower SMR counties (Fig 4).

## Discussion

This is the largest study to date, using electronic health records to explore leishmaniosis diagnosis in dogs under UK primary veterinary care. The study identified 976 dogs having a diagnosis of leishmaniosis from a study population of over

**Table 3. Descriptive statistics and mixed-effects univariable logistic regression (with clinic ID included as a random effect) results of association between demographic risk factors and having leishmaniosis in dogs in the UK in 2019 from VetCompass data.**

| | Cases (%) n = 915 | Controls (%) n = 38,812 | Odds Ratio (95%CI[1]) | Category p value | Variable p value |
|---|---|---|---|---|---|
| **Sex** | | | | | |
| Females | 382 (41.7) | 18,553 (47.8) | Reference | | <0.01 |
| Males | 532 (58.1) | 20,051 (51.7) | 1.29 (1.12-1.47) | <0.01 | |
| Missing | 1 (0.2) | 208 (0.5) | 0.22 (0.03-1.58) | 0.13 | |
| **Neutered** | | | | | |
| Yes | 707 (77.3) | 20,690 (53.3) | 2.89 (2.46-3.39) | <0.01 | |
| No | 208 (22.7) | 17,914 (46.2) | Reference | | <0.01 |
| Missing | 0 (0.0) | 208 (0.5) | | | |
| **Age category (years)** | | | | | |
| 1-2.99 | 200 (21.9) | 9,017 (23.2) | Reference | | <0.01 |
| 3.0-4.99 | 247 (27.0) | 7,710 (19.8) | 1.43 (1.18-1.73) | <0.01 | |
| 5.0-6.99 | 187 (20.4) | 6,663 (17.2) | 1.27 (1.04-1.56) | 0.02 | |
| 7.0-9.0 | 138 (15.1) | 5,729 (14.8) | 1.09 (0.87-1.36) | 0.47 | |
| >9 | 143 (15.6) | 9,682 (24.9) | 0.65 (0.52-0.81) | <0.01 | |
| Missing | 0 (0.0) | 11 (0.1) | | | |
| **Age (continuous in years) (median/IQR)** | 5.08 (IQR 3.32–7.71) | 5.74 (IQR 3.24–9.03) | 0.95 (0.94-0.97) | <0.01 | <0.01 |
| **Purebred status/coat length** | | | | | |
| Crossbreed | 508 (55.5) | 11,490 (29.6) | Reference | | <0.01 |
| Purebred short | 224 (24.5) | 14,708 (37.9) | 0.34 (0.29-0.40) | <0.01 | |
| Purebred medium | 164 (17.9) | 9,299 (24.0) | 0.38 (0.31-0.45) | <0.01 | |
| Purebred long | 15 (1.6) | 3,188 (8.2) | 0.11 (0.07-0.18) | <0.01 | |
| Missing | 4 (0.5) | 127 (0.3) | 0.69 (0.20–1.91) | 0.47 | |
| **Breed** | | | | | |
| Crossbreed | 524 (57.2) | 11,098 (28.6) | Reference | | <0.01 |
| Ibizan Hound | 6 (0.7) | 3 (0.008) | 48.0 (10.20-226.5) | <0.01 | |
| Brittany Spaniel | 24 (2.6) | 16 (0.04) | 43.3 (21.3-88.1) | <0.01 | |
| English Pointer | 25 (2.7) | 18 (0.05) | 32.8 (16.6-64.9) | <0.01 | |
| English Setter | 8 (0.9) | 20 (0.05) | 8.61 (3.49-21.2) | <0.01 | |
| German Pointer | 19 (2.0) | 97 (0.3) | 4.37 (2.57-7.43) | <0.01 | |
| Doberman Pinscher | 7 (0.8) | 85 (0.2) | 1.89 (0.85-4.21) | 0.12 | |
| Boxer | 17 (1.9) | 316 (0.8) | 1.13 (0.68-1.87) | 0.44 | |
| Greyhound | 11 (1.2) | 276 (0.7) | 0.83 (0.44-1.54) | 0.55 | |
| German Shepherd Dog | 29 (3.2) | 770 (2.0) | 0.77 (0.52-1.13) | 0.17 | |
| Beagle | 13 (1.4) | 385 (1.0) | 0.70 (0.40-1.24) | 0.22 | |
| Labrador Retriever | 29 (3.2) | 2,713 (7.0) | 0.21 (0.15-0.31) | <0.01 | |
| Yorkshire Terrier | 9 (1.0) | 1,037 (2.7) | 0.18 (0.09-0.35) | <0.01 | |
| Border Collie | 8 (0.9) | 1,139 (2.9) | 0.15 (0.07-0.30) | <0.01 | |
| Cocker Spaniel | 10 (1.1) | 1,691 (4.4) | 0.12 (0.06-0.23) | <0.01 | |
| Jack Russell Terrier | 9 (1.0) | 1,969 (5.1) | 0.09 (0.05-0.18) | <0.01 | |
| Staffordshire Bull Terrier | 6 (0.7) | 1,811 (4.7) | 0.07 (0.03-0.16) | <0.01 | |
| Other purebred | 157 (17.2) | 15,241 (39.2) | 0.20 (0.17-0.25) | <0.01 | |
| Missing | 4 (0.4) | 127 (0.3) | 0.64 (0.23-1.77) | 0.39 | |

[1]CI = confidence interval.

**Table 4. Descriptive statistics and mixed-effects univariable logistic regression (with clinic ID included as a random effect) results of association between geographic risk factors and having leishmaniosis in dogs in the UK in 2019 from VetCompass data.**

| | Cases (%) n=915 | Controls (%) n=38,812 | Odds Ratio(95%CI[1]) | Category p-value | Variable p-value |
|---|---|---|---|---|---|
| **Country of residence** | | | | | |
| England | 860 (94.0) | 33,938 (87.4) | reference | | 0.007 |
| Northern Ireland | 5 (0.5) | 611 (1.6) | 0.31 (0.12-0.82) | 0.02 | |
| Scotland | 30 (3.3) | 1,515 (3.9) | 0.78 (0.50-1.22) | 0.26 | |
| Wales | 19 (2.1) | 1,392 (3.6) | 0.52 (0.30-0.88) | 0.02 | |
| Missing | 1 (0.1%) | 1,356 (3.5) | 0.02 (0.04-0.19) | <0.01 | |
| **IMD5[2] category** | | | | | |
| 1 (most deprived) | 109 (11.9) | 7,077 (18.2) | 0.66 (0.50-0.89) | <0.01 | |
| 2 | 197 (21.5) | 8,692 (22.4) | reference | | <0.01 |
| 3 | 207 (22.6) | 7,755 (20.0) | 1.13 (0.88-1.45) | 0.34 | |
| 4 | 193 (21.1) | 7,211 (18.6) | 1.20 (0.93-1.55) | 0.17 | |
| 5 (least deprived) | 207 (22.6) | 6,721 (17.3) | 1.40 (1.09-1.80) | <0.01 | |
| Missing | 2 (0.2) | 1,356 (3.5) | 0.06 (0.01-0.26) | <0.01 | |
| **Urban/rural 4-category** | | | | | |
| Mixed urban/rural | 567 (62.0) | 20,719 (53.4) | reference | | <0.01 |
| Rural | 99 (10.8) | 4,019 (10.4) | 0.85 (0.64-1.13) | 0.27 | |
| Urban | 247 (27.0) | 12,606 (32.7) | 0.73 (0.61-0.89) | 0.01 | |
| Urban city/town | 0 (0.0) | 112 (0.3) | Not included | | |
| Missing | 2 (0.2) | 1,356 (3.2) | 0.05 (0.12-0.21) | <0.01 | |
| **Data source** | | | | | |
| Group 1 | 2 (0.2) | 31 (0.1) | 3.36 (0.47-24.2) | 0.23 | |
| Group 2 | 318 (34.8) | 11,348 (29.2) | 1.41 (1.14-1.75) | <0.01 | |
| Group 3 | 256 (28.0) | 12,134 (31.4) | reference | | <0.01 |
| Group 4 | 27 (3.0) | 742 (1.9) | 1.73 (1.07-2.79) | 0.02 | |
| Group 5 | 206 (22.4) | 7,085 (18.2) | 1.40 (1.11-1.77) | <0.01 | |
| Group 6 | 106 (11.6) | 6,029 (15.5) | 0.88 (0.66-1.17) | 0.40 | |
| Missing | 0 (0.0) | 1,443 (3.7) | | | |

[1] CI = Confidence interval.

[2] IMD = Indices of Multiple Deprivation.

2 million dogs in 2019. Crossbreeds were identified as having higher odds of leishmaniosis compared to purebreds when grouped together, although analysis on individual breeds revealed some individual breeds with very high odds of being a case. Attending a veterinary clinic outside England, in the least affluent or an urban area all had a reduced odds with being a case. Spatial clustering supported prior work, indicating that there was a higher risk of cases being found in southern England [10,19,38] with the highest SMRs being in Kent, Surrey, East Sussex and West Sussex.

The estimated 2019 prevalence estimate of leishmaniosis in dogs from this population was 0.0434%. It is acknowledged that the current results were estimated from case records of dogs presenting to clinicians at veterinary clinics participating in VetCompass rather than the whole of the UK. These clinicians were only likely to consider diagnostic testing for leishmaniosis with a reasonable level of suspicion of disease, as indicated by a history of originating from an endemic country and appropriate clinical signs, or in surveillance where imported dogs had tested positive pre-import. We would argue that this approach to testing would be likely to identify the majority of cases as sampling from the whole UK dog population, as there is little evidence of leishmaniosis reported in dogs that have not spent time out of the country (10,19,38). This prevalence estimate was slightly higher than the only previous UK prevalence estimate from 2016

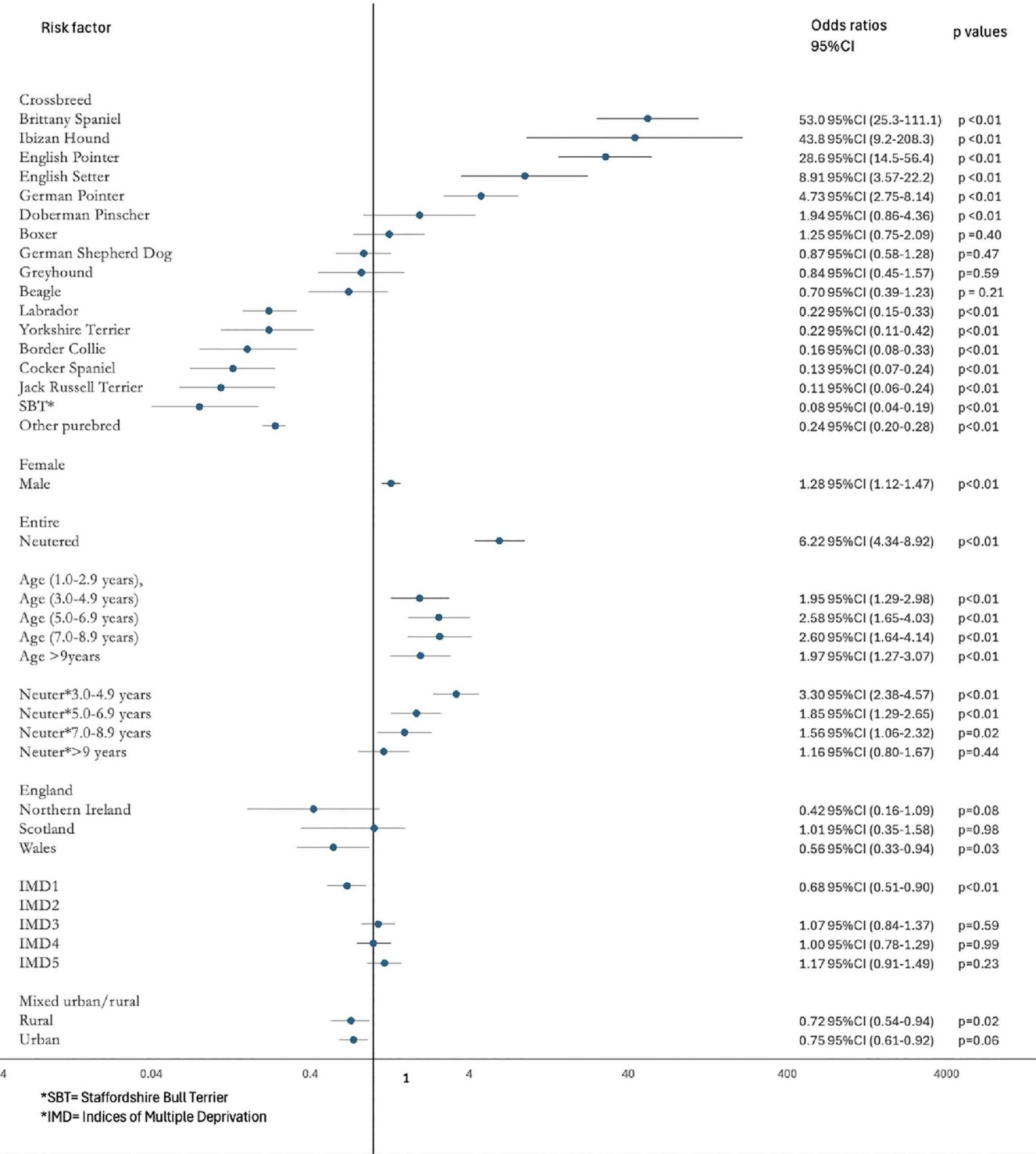

**Fig 1. Final mixed-effects multivariable logistic regression model results for risk factors associated with prevalence of having a leishmaniosis diagnosis in dogs in the UK in 2019.**

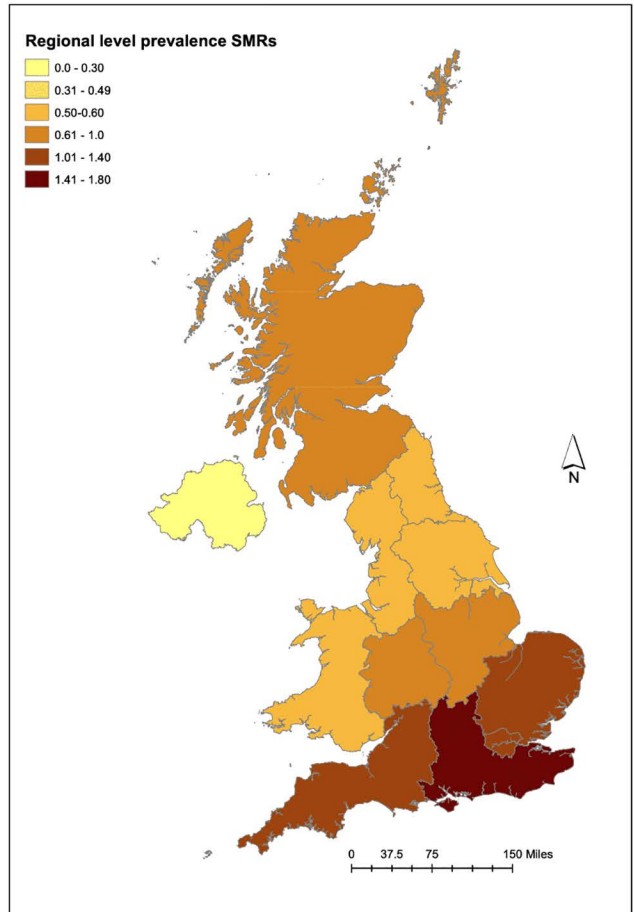
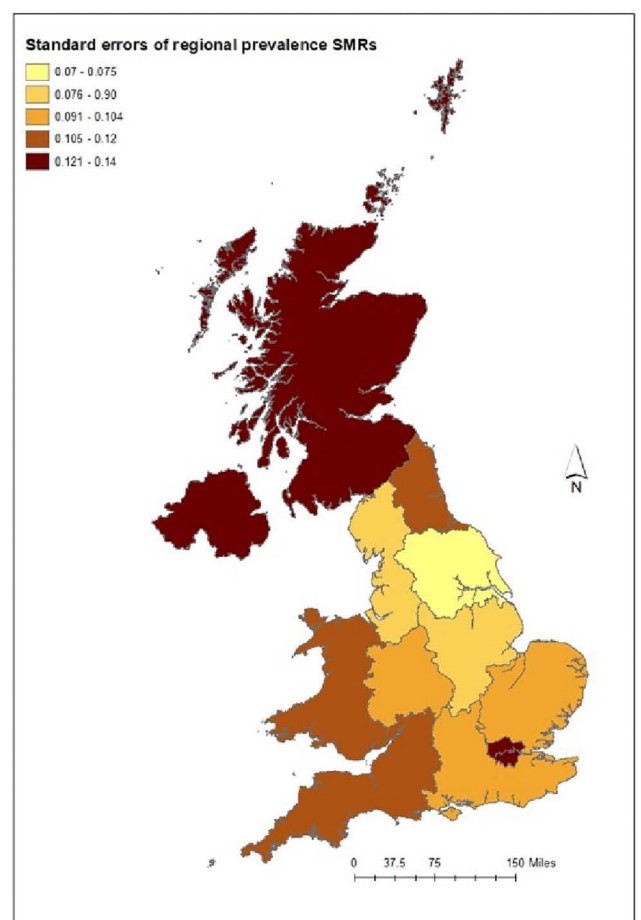

**Fig 2. Choropleth maps showing NUTS1 regional prevalence SMRs and standard errors of SMRs of having a leishmaniosis diagnosis in dogs in the UK in 2019.** (Contains National Statistic data © Crown copyright and database right [2024]).

(0.007–0.4%) [19]. The 2016 estimate was made from 38 cases taken from referral practice data and excluded dogs without clinical signs. Including subclinical dogs as cases may be important as these individuals may become clinical cases [18] and some studies report that subclinical cases could be a reservoir of the leishmania parasite for the sandfly vector to transmit disease [8,20].

The current study identified that 955 (97.8%) of the 976 cases had evidence of being imported or rescued from overseas, 20 (2.1%) had travelled from the UK and only one dog (0.01%) had no evidence of history of travel outside the UK. There were 189 (19.4%) dogs that had been reported as imported but no source country had been recorded. The proportion of cases having evidence of being imported/rescued from overseas is much higher than proportion reported in the most recent study from 2014 to 2022 looking at EHRs from practices on the Small Animal Surveillance Network (SAVSNET) which reported that from 368 cases, 189 (51.3%) had evidence of travel overseas and 92 of the cases (25%) had evidence in their history of being rescued from overseas or imported [38]. The current study gives further evidence that most of the leishmaniosis diagnosed dogs are from rescue or imported animals. Most imported or rescued cases had been diagnosed prior to arrival in the UK (581, 60.8%) which indicates that the new owners were aware their dog was infected with leishmaniosis despite this being a potentially serious infection. More education for prospective owners about the welfare and cost implications of owning a dog diagnosed with leishmaniosis may reduce this proportion.

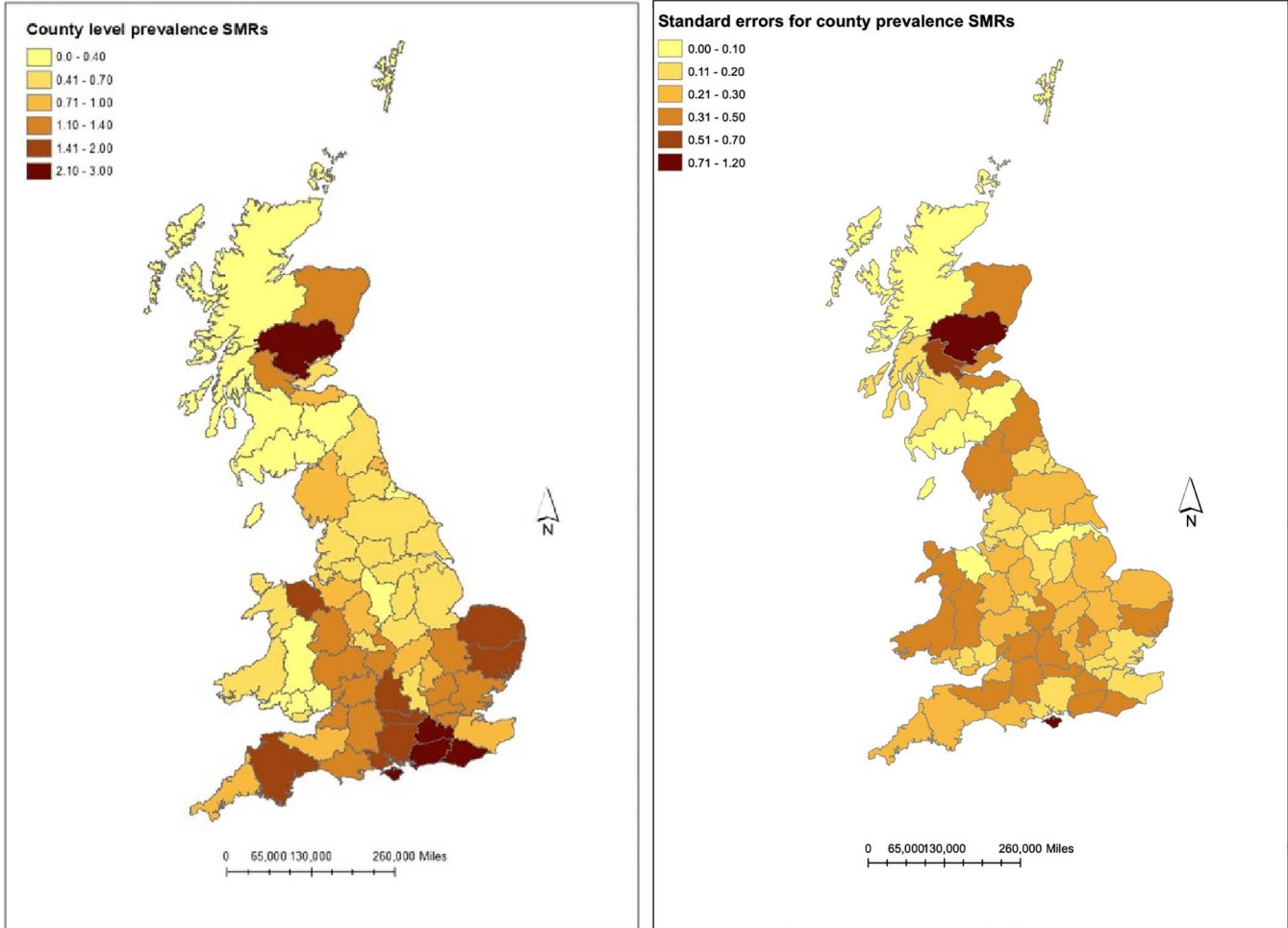

**Fig 3. Choropleth maps showing county prevalence SMRs and standard error of SMRs of having a leishmaniosis diagnosis in dogs in the UK in 2019.** (Contains National Statistic data © Crown copyright and database right [2024]).

Of the 395 dogs that were first diagnosed in the UK, 54 (13.7%) had tested negative in pre-import tests. This finding illustrates the difficulties of pre-import testing where sensitivity and specificity of serological tests are imperfect. Reported ELISA and IFAT sensitivity/specificity vary between 60 and 100% [23–28] so both false positive and negative results can occur. Additionally, infected animals may take up to five months or longer to show a significant serological response [25], therefore dogs may be negative on initial blood test but subsequently can become highly seropositive post-import. More studies into the effectiveness on testing protocols would be required before a mandatory testing scheme could be advised.

The current study found that being male was associated with increased odds of being a case. This aligns with some previous studies from Spain, Italy and the UK [30–32,38] although other studies from Italy, Morocco, Brazil, Portugal and Cyprus found no statistical difference between the sexes [33–37,41]. It has been postulated that male dogs may have higher odds of being a case in endemic countries due to increased roaming and therefore exposure to the sandfly vector [31] or that testosterone reduces the cell mediated immunity to the *Leishmania* parasite [42].

Being neutered was positively associated with being a case in this study which had been reported in the previous UK study [38]. The association of neuter status with being a leishmaniosis case has not been reported from studies in

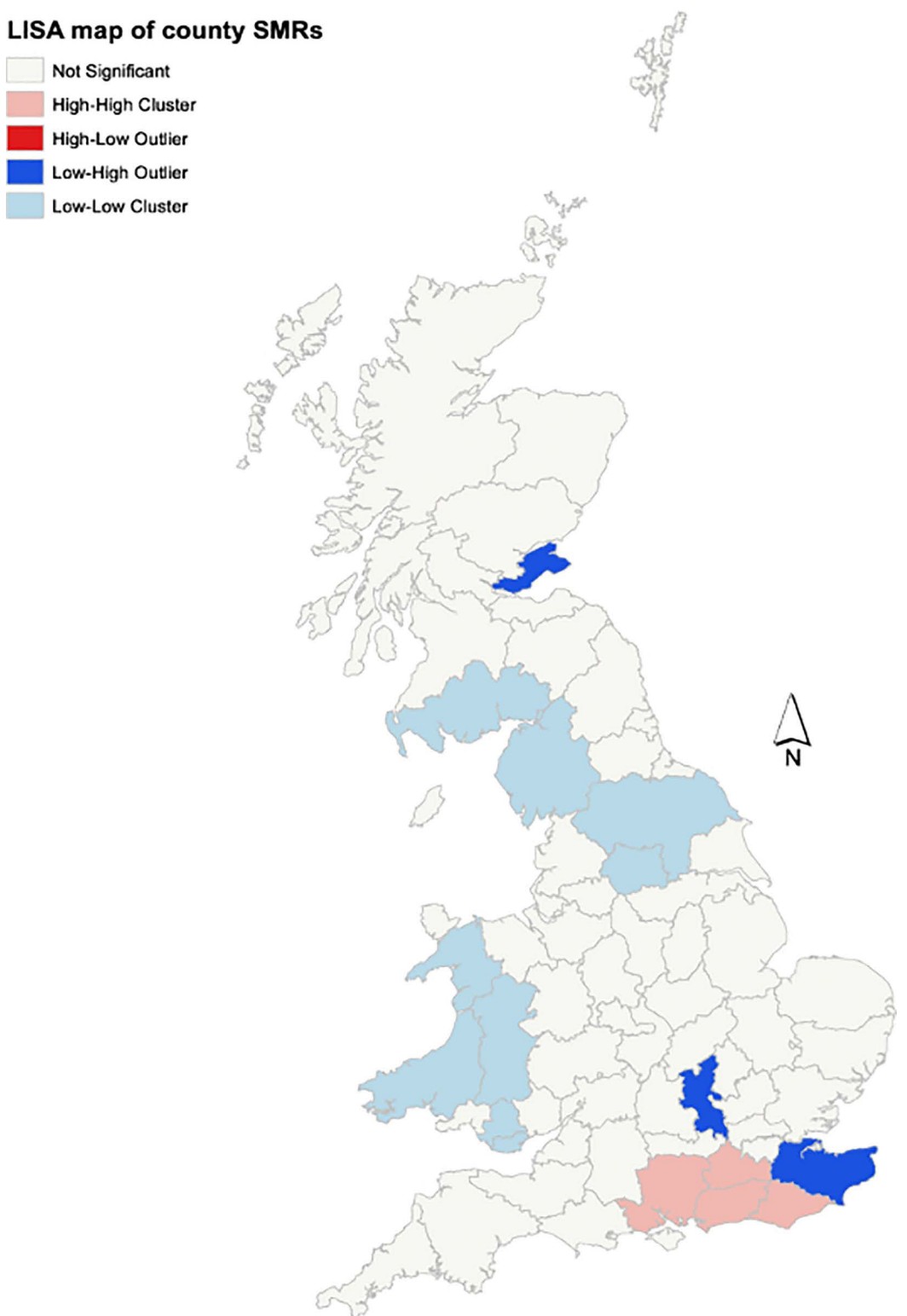

**Fig 4. LISA map of county prevalence SMRs of having a leishmaniosis diagnosis in dogs in the UK in 2019.** (Contains National Statistic data © Crown copyright and database right [2024]).

endemic countries and may reflect at least in part, active neutering policies for dogs that are being rehomed or imported to the UK. There was statistical evidence of an interaction between age and neutering, with the increased odds of having leishmaniosis observed with neutering reducing with increased age. There is no clear biological explanation for this finding, but many of the imported dogs were rehomed through charities, which would increasingly select for younger dogs that would automatically be neutered as part of the rehoming process. Older dogs that were not neutered were less likely to be rehomed through charities and therefore less likely to be a case.

In the current study, dogs older than three years were more likely to be a case than dogs under three years. This finding is mirrored in many previous studies from Portugal, Italy, Morocco, Brazil, and Cyprus [2,31,33–37,41] and is also reported from the previous UK study [38]. This is likely to reflect that in a chronic disease like leishmaniosis, the frequency will increase with age as more dogs with the infection survive. Additionally, it has been reported, that the time for seroconversion from exposure to the infected sandfly could take several months [12,25] so this would further delay diagnosis of infection. Further work looking at incident data may give more information regarding specific age susceptibility to becoming a diagnosed leishmaniosis case.

Crossbreeds had around four times the odds of being a case compared to most purebreds in the current study. This was contrary to other studies from Portugal, Italy, Morocco and Brazil [2,31,34,37]. where it was found crossbreeds had reduced odds of being a case compared to purebreds. This was thought to be due to a better innate immunity in crossbreed dogs [37]. The increased odds of having a leishmaniosis diagnosis in crossbreeds was reported in the most recent UK study [38] and may reflect that a higher proportion of cases in the UK would be rescued or imported crossbreeds whereas the breed distribution of controls would reflect the distribution of breeds presenting to UK practices. In a study from Spain, Boxers, GSD and Rottweilers have been reported to have increased risk of diagnosis [30]. The current study found that Boxers (OR 1.25, 95% CI 0.75–2.09) and GSD (OR 0.87, 95% CI 0.58–1.28) had approximately four time the odds of being a case compared to the other purebreds (OR 0.24, 95% CI 0.20–0.28). Whether these breeds are positively associated with having leishmaniosis compared to purebeds or are more common in endemic countries is unclear. Particularly high odds were observed in, Brittany Spaniels (OR 53.0, 95% CI 25.3–111.1)), Ibizan Hounds (OR 43.8, 95% CI 9.20–208.3), English Pointers (OR 28.6, 95% CI 14.5–56.4), English Setters (OR 8.91, 95% CI 3.57–22.2) and German Pointers (OR 4.73, 95% CI 2.75–8.14). This is contrary to a study from Spain showing that Ibizan Hounds have been found to be more resistant to developing leishmaniosis [43]. In the current study these high breed associations may reflect selection bias towards those breeds more likely to travel or be rescued/imported than those at increased risk of developing leishmaniosis. In this study nearly all the cases had been infected with leishmaniosis before they were imported. This means that all risk factor associations are of being imported as a case of canine leishmaniosis rather than becoming one.

Visiting a veterinary clinic in the least affluent area (IMD1) was associated with lower odds of being a case (OR 0.66, 95% CI 0.50–0.89). This may reflect that importation of a dog and/or confirmation of a diagnosis with laboratory tests is more likely with affluent owners. A previous study from Italy found a higher prevalence in owned dogs compared to kennel dogs and it was postulated that this was associated with increased testing in owned dogs [33].

This study supported the previous UK studies suggesting that dogs living in the south of England had increased odds of being a case [10,19,38]. The studies from 2009 and 2016 were descriptive studies and recorded number of cases [10,19] The most recent study from 2024 reported the proportion of positive samples from laboratory data [38]. This current study used SMRs and local spatial autocorrelation of having a leishmaniosis diagnosis to stengthen the statistical evidence of this association. This spatial distribution of cases may reflect greater affluence in this area which would make importation of dogs more likely and increase the likelihood of expensive diagnosis confirmation. It may also reflect an increased number of active rehoming charities in this region. Additionally, it is feasible that there is more travel of owners and their dogs to endemic areas from southern England although the current study found dogs travelling to Europe did not contribute hugely to the number of cases. A higher concentration of cases in southern England may also increase veterinarians' awareness of the disease and therefore could increase the likelihood of diagnosis.

A limitation common to all studies involving EHR, is that data are collected for different reasons than the purpose of the study, with potential for missing data on risk factors of interest [44]. Measurement bias involving dates of diagnosis and age estimates were especially likely when involving rescue dogs that are being re-homed by a charity. False postives cases were possible among those cases diagnosed pre-import on serology results alone and that remained on treatment throughout the study period without being re-tested as these may have just been exposed dogs. False negative cases were possible, as true cases can be challenging to diagnose and are reported to often present with non-specific signs [11]. Misclassification bias reduces accuracy in the measures of association and if non-differential may push odds ratios towards the null [29]. A limitation of the estimate of the prevalence reported is that this was not derived from a random sample of dogs from the whole population of dogs in the UK. We would argue that this was a relevant estimate for veterinary practice attending dogs as it relates to the diagnosis of leishmaniosis within a large cohort of veterinary practices across the UK.

Using the postcode of the veterinary practice as a proxy for the location of the dogs was a limitation for the spatial analysis. It is possible that owners could travel across county/region boundaries, from areas of different deprivation or development indices to the practice, causing a non-differential misclassification. This was considered a better option than the alternative option which was using the owner's partial postcode.

## Conclusions

The current study highlights a slight increase in the prevalence of leishmaniosis in the UK dog population in 2019 since the last prevalence estimate in 2016. Importation or rescuing of dogs from endemic areas appears to be the largest contributor to the prevalence of infected dogs. The odds of having leishmaniosis appear higher in male, neutered dogs greater than three years old and belonging to breeds more commonly imported. These findings can help veterinarians in the UK identify those dogs at greater risk of leishmaniosis, particularly in those with a history of importation from an endemic country. The south of England had the highest risk of canine leishmaniosis and its warmer climate and proximity to mainland Europe would also make it the most at risk to the incursion of the sandfly vector. Future ongoing surveillance targeted to this area would be sensible to check for the possibility of the sandfly transmission of leishmaniosis in the UK.

## Supporting information

**S1 Appendix. Import of dogs – information received from APHA access to information team.**
(PDF)

**S2 Appendix. Import of dogs – information received from APHA access to information team.**
(PDF)

## Author contributions

**Supervision:** Dave C. Brodbelt.

**Writing – original draft:** Simon C. Kent.

**Writing – review & editing:** Dan G. O'Neil, Kim B. Stevens, Laia Solano-Gallego.

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
