## [Decision Letter · Decision Letter 0]

4 Nov 2025

Dear Dr. Kent,

Thank you for submitting your manuscript to PLOS ONE. After careful consideration, we feel that it has merit but does not fully meet PLOS ONE’s publication criteria as it currently stands. Therefore, we invite you to submit a revised version of the manuscript that addresses the points raised during the review process.

We look forward to receiving your revised manuscript.

Kind regards,

Angela Monica Ionica, Ph.D.

Academic Editor

PLOS ONE

Journal Requirements:

2. Please upload a new copy of Figures 1, 2, 3, and 4 as the details are not clear. Please follow the link for more information:  https://journals.plos.org/plosone/s/figures

3. We note that Figures 2, 3, and 4 in your submission contain map images which may be copyrighted. All PLOS content is published under the Creative Commons Attribution License (CC BY 4.0), which means that the manuscript, images, and Supporting Information files will be freely available online, and any third party is permitted to access, download, copy, distribute, and use these materials in any way, even commercially, with proper attribution. For these reasons, we cannot publish previously copyrighted maps or satellite images created using proprietary data, such as Google software (Google Maps, Street View, and Earth). For more information, see our copyright guidelines: http://journals.plos.org/plosone/s/licenses-and-copyright.

1. You may seek permission from the original copyright holder of Figures 2, 3, and 4 to publish the content specifically under the CC BY 4.0 license. 

Reviewers' comments:

Reviewer's Responses to Questions

**Comments to the Author**

1. Is the manuscript technically sound, and do the data support the conclusions?

Reviewer #1: Yes

Reviewer #2: Yes

2. Has the statistical analysis been performed appropriately and rigorously?

Reviewer #1: Yes

Reviewer #2: Yes

3. Have the authors made all data underlying the findings in their manuscript fully available?

Reviewer #1: Yes

Reviewer #2: Yes

4. Is the manuscript presented in an intelligible fashion and written in standard English?

Reviewer #1: Yes

Reviewer #2: Yes

Reviewer #1: Dear Authors,

Thank you for submitting your manuscript titled “The frequency, risk factors and spatial distribution associated with having a diagnosis of leishmaniosis in dogs in the UK in 2019” to PLOS ONE. I would like to appreciate the importance of this study and the considerable effort invested in data extraction, validation, and analysis. Your work provides meaningful insights into the occurrence and determinants of canine leishmaniosis in a non-endemic region and contributes significantly to understanding the public health relevance of imported vector-borne infections.

The manuscript is well-structured, comprehensive, and uses appropriate statistical and spatial methods. However, several key clarifications and minor improvements are needed to ensure scientific transparency and reproducibility:

Suggestions for Improvements:

* Please specify the reference laboratories or diagnostic facilities used for IFAT, ELISA, and PCR assays, including the assay kits, positivity thresholds (cut-offs), and how results were interpreted when different methods were used. Also clarify how many cases were diagnosed using multiple diagnostic tests.

* Provide the criteria used to classify cases without confirmatory test results (approximately 25%). Indicate how these were validated and their diagnostic certainty.

* Specify if the conventional PCR or real-time PCR used for detection of disease. In case of qPCR whether it is inhouse method or any commercial kit used.

* Replace general use of “prevalence” with “diagnosed proportion” as the study results cannot be extrapolated as national prevalence due to non-random sampling.

* Discuss the potential sampling and reporting biases as imported dogs and rescue animals are more likely to be tested than the general dog population.

* Table 1 needs improvement by breaking down different variables. It is quite difficult to understand and interpret the table with multiple variables and information.

Your study provides valuable epidemiological evidence on canine leishmaniosis and could significantly aid veterinary surveillance and policy development once these clarifications are incorporated. I recommend minor revision, primarily to enhance clarity in terminology and diagnostic methodology with refined data interpretation.

Your efforts are appreciated and looking forward to receive your improved manuscript.

Best regards,

Muhammad Athar Abbas, DVM, PhD

Reviewer #2: This study aimed to identify the risk factors and spatial distribution of leishmaniosis in dogs in the UK. The authors analyzed a large number of dogs and presented the results obtained through detailed analyses. The findings confirmed that imported dogs from endemic countries or areas are the main sources of leishmaniosis. Although this result is expected and not surprising, the large-scale data analysis provides updated information on the epidemiology of leishmaniosis and contributes to disease control efforts in the UK. The authors should consider and discuss the following comments:

1. The definition of “leishmaniosis” should be clarified. The authors described clinical leishmaniosis in detail; however, the definition of subclinical leishmaniosis was not clearly stated. Is it equivalent to “seropositive” status in this study?

2. The authors concluded that certain breeds have very high odds of leishmaniosis. However, the distribution of imported dog breeds is not uniform and is biased according to breed popularity. This should be taken into account when making such a conclusion.

3. It would be helpful to include information on the distribution of sandflies in the UK to better discuss the transmission risk of leishmaniosis.

4. The term “leishmaniasis” is used in line 408, whereas “leishmaniosis” is used elsewhere. The terminology should be standardized according to the journal’s style.

5. The phrase “leishmaniasis infection” in line 408 should be revised, as “leishmaniasis” refers to the clinical condition itself.

6. Tables should be more organized.

**Do you want your identity to be public for this peer review?** For information about this choice, including consent withdrawal, please see our Privacy Policy

Reviewer #1: **Yes: ** Dr. Muhammad Athar Abbas (DVM, PhD)

Reviewer #2: No

---

## [Author Response · Author response to Decision Letter 1]

21 Dec 2025

PLEASE SEE REBUTTAL LETTER. BUT COPIED HERE:

The authors of this paper thank the academic editor and reviewers for their detailed, considered feedback. This letter includes our responses which will correspond to the track changes on the revised manuscript. Reviewer comments are shown with a preceding ‘C’ below, followed by any author responses with preceding ‘R’ below. Any revised wording is then given along with the line numbers. The lines quoted refer the tracked version of the re-submission.

Academic Editor

Comments to author

C: Please ensure that your manuscript meets PLOS ONE's style requirements, including those for file naming.

R: File names have been changed as per PLOS ONE style templates.

Comments to author

C: Please upload a new copy of Figures 1, 2, 3, and 4 as the details are not clear. Please follow the link for more information.

R: New figures have been constructed at a higher resolution to improve legibility.

Comments to author

C: We note that Figures 2, 3, and 4 in your submission contain map images which may be copyrighted. We require you to either (1) present written permission from the copyright holder to publish these figures specifically under the CC BY 4.0 license, or (2) remove the figures from your submission:

R: Figures 2,3 and 4 are original maps created by the first author as mentioned in lines 245-257 of the manuscript and therefore are not copyrighted.

Reviewer 1:

Comments to author

C: I would like to appreciate the importance of this study and the considerable effort invested in data extraction, validation, and analysis. Your work provides meaningful insights into the occurrence and determinants of canine leishmaniosis in a non-endemic region and contributes significantly to understanding the public health relevance of imported vector-borne infections.

R: The authors thank the reviewer for their feedback.

Comments to author

C: Please specify the reference laboratories or diagnostic facilities used for IFAT, ELISA, and PCR assays, including the assay kits, positivity thresholds (cut-offs), and how results were interpreted when different methods were used. Also clarify how many cases were diagnosed using multiple diagnostic tests.

R: Unfortunately, the original data are from clinical records from veterinary clinics within VetCompass and do not consistently hold information regarding which laboratories had been used. The number of cases that were diagnosed with multiple tests were not recorded in this study.

In the Methods lines 140-146 have been updated to” The positive diagnosis was confirmed by a commercial UK laboratory although the specific laboratory or diagnostic facilities where each test was performed were not available to the study. The results were interpreted according to that individual laboratory’s reference range. Results were recorded as either positive or negative as numerical values for each result could not be compared between laboratories. The number of positive cases that were confirmed with more than one test result was not extracted for this study.”

Comments to author

C: Provide the criteria used to classify cases without confirmatory test results (approximately 25%). Indicate how these were validated and their diagnostic certainty.

R: Lines 157-159 have been updated to “Dogs diagnosed with leishmaniosis pre-import, that remained on treatment without further confirmatory laboratory tests after arrival in the UK, were additionally classified as cases.”

The diagnostic certainty of pre-import tests could not be assessed. We have now acknowledged this in the limitations section of the discussion in lines 451-453 “False postives cases were possible among those cases diagnosed pre-import on serology results alone and that remained on treatment throughout the study period without being re-tested as these may have just been exposed dogs.”

Comments to author

C: Specify if the conventional PCR or real-time PCR used for detection of disease. In case of qPCR whether it is inhouse method or any commercial kit used.

R: Line 150 has been updated to specify that both PCR and qPCR have been used, as different laboratories at different times in the study period have used varying methodologies. Lines 140-142 has been updated to specify that diagnosis was confirmed in commercial laboratories rather than using inhouse methods. “The positive diagnosis was confirmed by a commercial UK laboratory although the specific laboratory or diagnostic facilities where each test was performed were not available to the study.”

Comments to author

C: Replace general use of “prevalence” with “diagnosed proportion” as the study results cannot be extrapolated as national prevalence due to non-random sampling .

R: We thank the reviewer for this valid point. We would prefer to retain some reference to prevalence as this relates to the clinical diagnosis within this large cohort of veterinary practices across the UK. However, we try to acknowledge the uncertain representativeness by now terming it estimated prevalence and qualify this further in the Abstract (lines 15-16) “The estimated prevalence of leishmaniosis within dogs under care at VetCompass participating practices was calculated”. and Discussion (lines 390-393) “The estimated 2019 prevalence estimate of leishmaniosis in dogs from this population was 0.0434%. It is acknowledged that this was estimated from case records of dogs presenting to clinicians at veterinary clinics participating in VetCompass rather than the whole of the UK.”

Comments to author

C: Discuss the potential sampling and reporting biases as imported dogs and rescue animals are more likely to be tested than the general dog population.

R: Thank you for this further point. Currently leishmaniosis is not considered endemic in the UK and there have been very few cases in dogs that haven’t been imported or travelled outside of the UK. It is a fair point that the only dogs likely to be tested are those with a history of importation or travel out of the UK. We have updated the discussion in lines 392-400 to acknowledge this non random sampling. In this updated discussion we have argued why we believe this will not affect the number of cases.

“It is acknowledged that the current results were estimated from case records of dogs presenting to clinicians at veterinary clinics participating in VetCompass rather than the whole of the UK. These clinicians were only likely to consider diagnostic testing for leishmaniosis with a reasonable level of suspicion of disease, as indicated by a history of originating from an endemic country and appropriate clinical signs or in surveillance where imported dogs had tested positive pre-import. We would argue that this approach to testing would be likely to identify the majority of cases as sampling from the whole UK dog population, as there is little evidence of leishmaniosis reported in dogs that have not spent time out of the country (10,19,38).”

We have updated the limitations section of the discussion at lines 503-506 to “A limitation of the estimate of the prevalence reported is that this was not derived from a random sample of dogs from the whole population of dogs in the UK. We would argue that this was a relevant estimate for veterinary practice attending dogs as it relates to the diagnosis of leishmaniosis within a large cohort of veterinary practices across the UK.”

Comments to author

C: Table 1 needs improvement by breaking down different variables. It is quite difficult to understand and interpret the table with multiple variables and information.

R: Thank you. Table 1 has been split into 2 separate tables to improve the clarity and hopefully make it easier for the reader to interpret the results

Reviewer 2

Comments to author

C: This study aimed to identify the risk factors and spatial distribution of leishmaniosis in dogs in the UK. The authors analyzed a large number of dogs and presented the results obtained through detailed analyses. The findings confirmed that imported dogs from endemic countries or areas are the main sources of leishmaniosis. Although this result is expected and not surprising, the large-scale data analysis provides updated information on the epidemiology of leishmaniosis and contributes to disease control efforts in the UK.

R: The authors thank the reviewer for their feedback.

Comments to author

C: The definition of “leishmaniosis” should be clarified. The authors described clinical leishmaniosis in detail; however, the definition of subclinical leishmaniosis was not clearly stated. Is it equivalent to “seropositive” status in this study?

R: This has been updated in lines 154-155 to “Subclinical leishmaniosis cases were defined as dogs that remained free of clinical signs but had persistent high seropositivity or presence of parasite in laboratory tests”.

Comments to author

C: The authors concluded that certain breeds have very high odds of leishmaniosis. However, the distribution of imported dog breeds is not uniform and is biased according to breed popularity. This should be taken into account when making such a conclusion.

R: The authors agree with the reviewer on this point and has updated the abstract in lines 35-37 “The breeds with very high odds for leishmaniosis may reflect selection bias towards breeds more likely to be imported, rescued or travelled from endemic areas rather than high intrinsic risk in these breeds towards leishmaniosis.”

Additionally, this has been updated in the discussion in lines 465-474 “Particularly high odds were observed in, Brittany Spaniels (OR 53.0, 95% CI 25.3-111.1) ), Ibizan Hounds (OR 43.8, 95% CI 9.20-208.3), English Pointers (OR 28.6, 95% CI 14.5-56.4), English Setters (OR 8.91, 95% CI 3.57-22.2) and German Pointers (OR 4.73, 95% CI 2.75-8.14). This is contrary to a study from Spain showing that Ibizan Hounds have been found to be more resistant to developing leishmaniosis [44]. In the current study these high breed associations may reflect selection bias towards those breeds more likely to travel or be rescued/imported than those at increased risk of developing leishmaniosis.”

Comments to author

C: It would be helpful to include information on the distribution of sandflies in the UK to better discuss the transmission risk of leishmaniosis.

R: This has been updated in lines 50-52 “Transmission risk in the UK remains very low because no incursions of the sandfly vector have been recorded into the UK.”

Comments to author

C: The term “leishmaniasis” is used in line 408, whereas “leishmaniosis” is used elsewhere. The terminology should be standardized according.

R: This error has been corrected.

Comments to author

C: The phrase “leishmaniasis infection” in line 408 should be revised, as “leishmaniasis” refers to the clinical condition itself.

R: This error has been corrected.

Comments to author

C: Tables should be more organized.

R: Table 1 has been re-organised into two tables to hopefully make it easier for the reader to interpret.

---

## [Decision Letter · Decision Letter 1]

11 Jan 2026

The frequency, risk factors and spatial distribution associated with having a diagnosis of leishmaniosis in dogs in the UK in 2019

PONE-D-25-49806R1

Dear Dr. Kent,

We’re pleased to inform you that your manuscript has been judged scientifically suitable for publication and will be formally accepted for publication once it meets all outstanding technical requirements.

Kind regards,

Angela Monica Ionica, Ph.D.

Academic Editor

PLOS One

Additional Editor Comments (optional):

Reviewers' comments:

Reviewer's Responses to Questions

**Comments to the Author**

Reviewer #2: All comments have been addressed

2. Is the manuscript technically sound, and do the data support the conclusions?

Reviewer #2: Yes

3. Has the statistical analysis been performed appropriately and rigorously?

Reviewer #2: Yes

4. Have the authors made all data underlying the findings in their manuscript fully available?

Reviewer #2: Yes

5. Is the manuscript presented in an intelligible fashion and written in standard English?

Reviewer #2: Yes

Reviewer #2: The manuscript has been properly revised. I have confirmed that all the necessary changes have been made, and I believe it is now ready for publication.

**Do you want your identity to be public for this peer review?** For information about this choice, including consent withdrawal, please see our Privacy Policy

Reviewer #2: No

---

## [Editor Report · Acceptance letter]

PONE-D-25-49806R1

PLOS One

Dear Dr. Kent,

I'm pleased to inform you that your manuscript has been deemed suitable for publication in PLOS One. Congratulations! Your manuscript is now being handed over to our production team.

Kind regards,

on behalf of

Dr. Angela Monica Ionica

Academic Editor

PLOS One